# GRAPH DECISION TRANSFORMER

## ABSTRACT

Offline Reinforcement Learning (RL) is a challenging task, whose objective is to learn policies from static trajectory data without interacting with the environment. Recently, offline RL has been viewed as a sequence modeling problem, where an agent generates a sequence of subsequent actions based on a set of static transition experiences. However, existing approaches that use Transformers to attend to all tokens plainly may dilute the truly-essential relation priors due to information overload. In this paper, we propose the Graph Decision Transformer (GDT), a novel offline RL approach that models the input sequence into a causal graph to capture potential dependencies between fundamentally different concepts and facilitate temporal and causal relationship learning. GDT uses a Graph Transformer to process the graph inputs with relation-enhanced mechanisms, and an optional Patch Transformer to handle fine-grained spatial information in visual tasks. Our experiments show that GDT matches or surpasses the performance of state-of-the-art offline RL methods on image-based Atari and D4RL benchmarks.

## 1 INTRODUCTION

Reinforcement Learning (RL) is inherently a sequential process where an agent observes a state from the environment, takes an action, observes the next state, and receives a reward. To model RL problems, Markov Decision Processes (MDPs) have been widely employed, where an action is taken solely based on the current state, which is assumed to encapsulate the entire history. Online RL algorithms (Mnih et al., 2015) use the temporal difference (TD) learning to train agents by interacting with the environment, but this can be prohibitively expensive in real-world settings. Offline RL (Levine et al., 2020), on the other hand, seeks to overcome this limitation by learning policies from a pre-collected dataset, without the need to interact with the environment. This approach makes RL training more practical for real-world scenarios and has therefore garnered significant attention.

Recent advances (Chen et al., 2021; Janner et al., 2021) in offline RL have taken a new perspective on the problem, departing from conventional methods that concentrate on learning value functions (Riedmiller, 2005; Kostrikov et al., 2021a) or policy gradients (Precup, 2000; Fujimoto & Gu, 2021). Instead, the problem is viewed as a generic sequence modeling task, where past experiences consisting of state-action-reward triplets are input to Transformer (Vaswani et al., 2017). The model generates a sequence of action predictions using a goal-conditioned policy, effectively converting offline RL to a supervised learning problem. This approach relaxes the MDP assumption by considering multiple historical steps to predict an action, allowing the model to be capable of handling long sequences and avoid stability issues associated with bootstrapping (Srivastava et al., 2019; Kumar et al., 2019c). Furthermore, this framework unifies multiple components in offline RL, such as estimating the behavior policy and predictive dynamics modeling, into a single sequence model, resulting in superior performance.

However, this approach faces three major issues. Firstly, states and actions represent fundamentally different concepts (Villaflor et al., 2022). While the agent has complete control over its action sequences, the resulting state transitions are often influenced by external factors. Thus, modeling states and actions as a single sequence may indiscriminate the effects of the policy and world dynamics on the return, which can lead to overly optimistic behavior. Secondly, in RL problems, the adjacent states, actions, and rewards are typically strongly connected due to their potential causal relationships. Specifically, the state observed at a given time step is a function of the previous state and action, and the action taken at that time step influences the subsequent state and reward. Simply applying Transformer to attend to all tokens without considering the underlying Markovian relationship can result in an overabundance of information, hindering the learning process in accurately

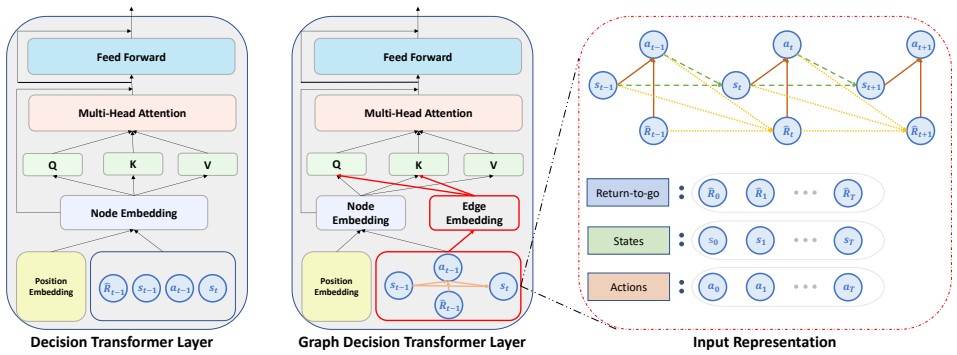

Figure 1: Comparison of GDT and DT. GDT employs additional edge embeddings and node embeddings to obtain Q and K, while using only node embeddings to obtain V. The corresponding input graph is depicted on the right side, the directed edge connecting $s_{t-1}$ to $s_t$ indicates the influence of $s_{t-1}$ on $s_t$, and similar relationships apply to other directed edges.

capturing essential relation priors and handling long-term sequences of dependencies from scratch. Finally, tokenizing image states as-a-whole using convolutional neural networks (CNNs) can hinder the ability of Transformers to gather fine-grained spatial relations. This loss of information can be especially critical in visual RL tasks that require detailed knowledge of regions-of-interest. Therefore, it is necessary to find a more effective way to represent states and actions separately while still preserving their intrinsic relationships, and to incorporate the Markovian property and spatial relations in the modeling process.

To alleviate such issues, we propose a novel approach, namely Graph Decision Transformer (GDT), which involves transforming the input sequence into a causal graph structure. The Graph Representation explicitly incorporates the potential dependencies between adjacent states, actions, and rewards, thereby better capturing the Markovian property of the input and differentiating the impact of different tokens. To process the input graph, we utilize the Graph Transformer to effectively handle long-term dependencies that may be present in non-Markovian environments. To gather fine-grained spatial information, we incorporate an optional Patch Transformer to encode image-based states as patches similar to ViT (Dosovitskiy et al., 2020), which helps with action prediction and reduces the learning burden of the Graph Transformer. Our experimental evaluations conducted in Atari and D4RL benchmark environments provide empirical evidence to support the advantages of utilizing a causal graph representation as input to the Graph Transformer in RL tasks. The proposed GDT method achieves state-of-the-art performance in several benchmark environments and outperforms most existing offline RL methods without incurring additional computational overhead.

In summary, our main contributions are four-fold:

• We propose a novel approach named GDT, that transforms input sequences into graph structures to better capture potential dependencies between adjacent states, actions, and rewards and differentiate the impact of these different tokens.

• We utilize the Graph Transformer to process the input graph, which can effectively handle long-term dependencies in the original sequence that may be present in non-Markovian environments.

• We incorporate an optional Patch Transformer to encode image-based states as patches to gather fine-grained spatial information crucial for visual input environments.

• We extensively evaluate GDT on Atari and D4RL benchmark environments, demonstrating its superior performance compared to existing offline RL methods.

## 2  RELATED WORK

**Offline RL.**  Offline RL has recently gained significant attention as an alternative paradigm, where agents extract return-maximizing policies from fixed, limited datasets composed of trajectory roll-outs from arbitrary policies (Levine et al., 2020). These datasets, referred to as static datasets, are formally defined as $\mathcal{D} = \{(s_t, a_t, s_{t+1}, r_t)_i\}$, where $i$ is the index, the actions and states are generated by the behavior policy $(s_t, a_t) \sim d^{\pi_\beta}(\cdot)$, and the next states and rewards are determined by

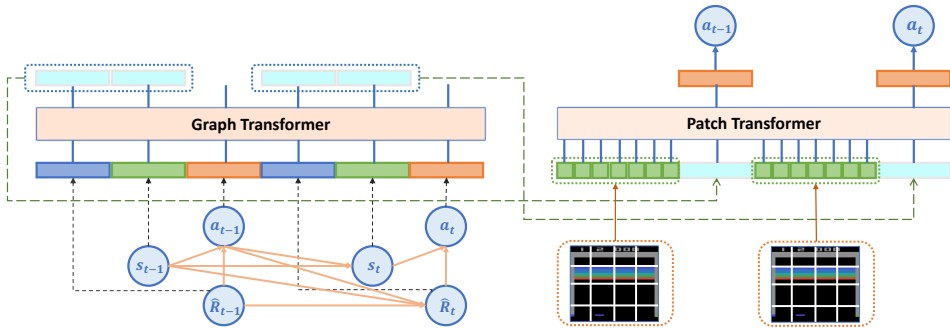

Figure 2: The proposed model comprises three main components: the Graph Representation, the Graph Transformer, and an optional Patch Transformer. When employing the direct output of the Graph Transformer for action prediction, the resultant model is denoted as GDT, representing the left half of the depicted figure. Alternatively, if the output of the Graph Transformer undergoes additional processing by the Patch Transformer, the resulting model is identified as GDT-plus, encompassing the entire figure.

the dynamics $(s_{t+1}, r_t) \sim (T(\cdot|s_t, a_t), r(s_t, a_t))$. Deploying off-policy RL algorithms directly in the offline setting is hindered by the distributional shift problem, which can result in a significant performance drop, as demonstrated in prior research (Fujimoto et al., 2019b). To mitigate this issue, model-free algorithms aim to either constrain the action space of policy (Kumar et al., 2019b; Siegel et al., 2020) or incorporate pessimism to value function (Kumar et al., 2020; Fujimoto et al., 2019b). Conversely, model-based algorithms simulate the actual environment to generate more data for policy training (Kidambi et al., 2020; Yu et al., 2020). In this work, we propose a novel approach that avoids learning the dynamics model explicitly and directly generates the next action with the help of the Graph Transformer, enabling better generalization and transfer (Ramesh et al., 2021).

**RL to Sequence Modeling.** RL has recently garnered considerable interest as a sequence modeling task, particularly with the application of Transformer-based decision models (Hu et al., 2022). The task is to predict a sequence of next actions given a sequence of recent experiences, including state-action-reward triplets. This approach can be trained in a supervised learning fashion, making it more amenable to offline RL and imitation learning settings. Several studies (Wang et al., 2022; Furuta et al., 2021; Zhang et al., 2023; Lee et al., 2022) have explored the use of Transformers in RL under the sequence modeling pattern. For example, Chen et al. (2021) train a Transformer as a model-free context-conditioned policy, while Janner et al. (2021) bring out the capability of the sequence model by predicting states, actions, and rewards and employing beam search. Zheng et al. (2022) further fine-tune the Transformer by adapting this formulation to online settings. Shang et al. (2022) explicitly model StAR-representations to introduce a Markovian-like inductive bias to improve long-term modeling. In this work, we propose a graph sequence modeling approach to RL, which explicitly introduces the Markovian property to the representations. Our proposed GDT method outperforms several state-of-the-art non-Transformer offline RL and imitation learning algorithms on Atari and D4RL benchmarks, demonstrating the advantage of incorporating graph structures in sequence modeling for RL tasks and the effectiveness of our proposed framework.

**RL with Graph.** In recent years, the integration of graph neural networks (GNNs) (Cai & Lam, 2020; Yun et al., 2019) with RL has attracted considerable attention for graph-structured environments (Munikoti et al., 2022). Specifically, GNNs can be combined with RL to address sequential decision-making problems on graphs. Existing research primarily focuses on using deep RL to improve GNNs for diverse purposes, such as neural architecture search (NAS) (Zhou et al., 2019), enhancing the interpretability of GNN predictions (Shan et al., 2021), and designing adversarial examples for GNNs (Dai et al., 2018; Sun et al., 2020). Alternatively, GNNs can be utilized to solve relational RL problems, such as those involving different agents in a multi-agent deep RL (MADRL) framework (Shen et al., 2021; Böhmer et al., 2020; Zhang et al., 2021), and different tasks in a multi-task deep RL (MTDRL) framework (Wang et al., 2018; Huang et al., 2020; Battaglia et al., 2018; Huang et al., 2022). Despite the growing interest in this field, there is currently a lack of research on

utilizing a Markovian dependency graph as input to GNNs for action prediction. Such an approach has strong potential due to the causal relationships between the constructed graph and its ability to be employed in various RL environments. This article will explore this approach in detail, offering a novel contribution to the field. Additionally, we will highlight the advantages of using GDT for action prediction in comparison to existing state-of-the-art offline RL algorithms.

## 3 METHODOLOGY

The proposed approach leverages both graph and sequence modeling techniques to create a deep learning-based model for offline RL tasks. The model consists of three main components: the Graph Representation, the Graph Transformer, and an optional Patch Transformer, as shown in Figure 2. Specifically, the Graph Representation is used to represent the input sequence as a graph with a causal relationship, thereby better capturing the Markovian property of the input and differentiating the impact of different tokens. The Graph Transformer then processes the graph inputs using the relation-enhanced mechanism, which allows the model to acquire long-term dependencies and model the interactions between different time steps of the graph tokens given reasonable causal relationships. The optional Patch Transformer is introduced to gather the fine-grained spatial information in the input, which is particularly important in visual tasks such as the Atari benchmark.

The proposed approach offers several advantages. Firstly, it can effectively acquire the intricate dependencies and interactions between different time steps in the input sequence, making it well-suited for RL tasks. Secondly, it encodes the sequence as a causal graph, which explicitly incorporates potential dependencies between adjacent tokens, thereby explicitly introducing the Markovian bias into the learning process and avoiding homogenizing all tokens. Finally, it can accurately gather fine-grained spatial information and integrate it into action prediction, leading to improved performance. In summary, our proposed approach offers a powerful and flexible solution for RL tasks. In the subsequent sections, we present an in-depth explanation of each component comprising our proposed approach. This involves elucidating its input and output mechanisms while delving into the underlying design principles that guide its construction.

### 3.1 GRAPH REPRESENTATION

Our approach constructs an input graph from trajectory elements, capturing their temporal and causal relationships. Each trajectory element corresponds to a node in the graph, with edges connecting them based on dependencies. Specifically, we define the directed graph $\mathcal{G} = \langle \mathcal{V}, \mathcal{E} \rangle$, where each node $v_i \in \mathcal{V}$ falls into one of three categories: returns-to-go $\hat{R}_t = \sum_{t'=t}^{T} r_{t'}$, states $s_t$, and actions $a_t$, as illustrated in Figure 1. Edges, denoted as $e_{i \to j} \in \mathcal{E}$, are introduced based on the dependencies between pairs of nodes. For instance, the selection of action $a_t$ relies on the current state $s_t$ and the returns-to-go $\hat{R}_t$. The generation of state $s_t$ depends on the previous state $s_{t-1}$ and action $a_{t-1}$. Lastly, the returns-to-go $\hat{R}_t$ is jointly determined by the previous returns-to-go $\hat{R}_{t-1}$, the state $s_{t-1}$, and the action $a_{t-1}$. The detailed design principles are outlined in Appendix A.

The MDP is a framework where an agent is asked to make a decision based on the current state $s_t$; then, the environment responds to the action made by the agent and transitions the state to the next state $s_{t+1}$ with a reward $r_t$. We omit the agent and environment entities and represent these dependencies as directed edges to effectively analyze the causal associations among distinct tokens. Through this approach, we explicitly embed the Markovian relationship bias into the input representation and distinguish the effect of distinct tokens, avoiding the overabundance of information that may prevent the model from accurately capturing essential relation priors.

### 3.2 GRAPH TRANSFORMER

Our approach employs auto-regressive modeling to predict current action using the Graph Representation as input to the Graph Transformer. The Graph Transformer plays a central role in extracting hidden features from the graph. In some scenarios, such as Atari games, past information also plays a critical role, thus we employ a global attention mechanism, allowing each node $v_i$ to observe not only its directly connected nodes $\{v_j | e_{j \to i} = 1\}$ but also all nodes preceding the current moment $\{v_j | j < i\}$. In the vanilla multi-head attention, the attention score between the element $v_i$ and $v_j$

can be formulated as the dot-product between their query vector and key vector, respectively:

$$s_{ij} = f(v_i, v_j) = v_i W_q^T W_k v_j. \tag{1}$$

Let us consider the score $s_{ij}$ as implicit information pertaining to the edge $e_{j \to i}$ (node $v_j$ influences node $v_i$). To enhance the previously calculated implicit attention score $s_{ij}$, we now try to incorporate available edge information for the specific edge $e_{j \to i}$. It is done by simply adding the node-relation interactions into the attention calculation, referred to as the relation-enhanced mechanism, as illustrated in Figure 1. The equation for computing the attention scores is shown below:

$$\begin{aligned} s_{ij} &= g(v_i, v_j, r_{i \to j}, r_{j \to i}) \\ &= (v_i + r_{i \to j}) W_q^T W_k (v_j + r_{j \to i}), \end{aligned} \tag{2}$$

where $r_{* \to *}$ is learned through an embedding layer that takes the adjacency matrix as input. We concurrently take into account both opposing directed edges between two nodes and adjust the query and key embeddings in accordance with the direction of the specific edge being examined. The incorporation of relation representation enables the model to take into account the plausible causal relationships, which relieves the burden on the Graph Transformer of learning potential long-term dependencies among node vectors at different time steps.

The input to the $l$-th layer of the Graph Transformer is a token graph. For the sake of clarity and simplicity, we represent the pruned graph as $\overline{G}$, which is utilized to generate the following sequence:

$$\overline{G}_{\text{in}}^l = \{\hat{R}_0^{l-1}, s_0^{l-1}, a_0^{l-1}, \ldots, \hat{R}_T^{l-1}, s_T^{l-1}, a_T^{l-1}\}. \tag{3}$$

Each token graph is transformed to $\{g_t^l\}_{t=0}^T$ by a Graph Transformer layer:

$$\begin{aligned} G_{\text{out}}^l &= F_{\text{graph}}^l(G_{\text{in}}^l) = F_{\text{graph}}^l(G_{\text{out}}^{l-1}), \\ g_t^l &:= \text{FC}(\overline{G}_{\text{out}}^l[1 + 3t], \overline{G}_{\text{out}}^l[3t]). \end{aligned} \tag{4}$$

As shown in Figure 1, the action $a_t$ is determined by both $\hat{R}_t$ and $s_t$. Thus, the feature vector $g_t^l$ is obtained by concatenating the two inputs and fed into a fully connected layer (with indexing starting from 0). The feature $g_t^l$ extracted from the $l$-th layer of the Graph Transformer can be directly used to predict the action $\hat{a}_t = \phi(g_t^l)$, or be further processed by the subsequent Patch Transformer.

### 3.3 PATCH TRANSFORMER

We introduce an additional Patch Transformer to assist with action prediction and reduce the learning burden of the Graph Transformer. The Patch Transformer adopts the conventional Transformer layer design from Vaswani et al. (2017) and is incorporated to gather fine-grained spatial information that is crucial for visual input environments. The initial layer of the Patch Transformer takes a collection of image patches and $g_t^0$ as inputs:

$$Y_{\text{in},t}^0 = \{z_{s_t^1}, z_{s_t^2}, \ldots, z_{s_t^n}, g_t^0\}, \tag{5}$$

where $n$ is the number of image patches, and the feature vector $g_t$ is positioned after state patches $\{z_{s_t^i}\}_{i=1}^n$, which enables $g_t$ to attend to all spatial information. We have $T$ groups of such token representations, which are simultaneously processed by the Patch Transformer:

$$\begin{aligned} Y_{\text{in}}^0 &= \overset{T}{\underset{t=0}{\|}} Y_{\text{in},t}^0, \\ Y_{\text{out}}^0 &= F_{\text{patch}}^0(Y_{\text{in}}^0), \end{aligned} \tag{6}$$

where $\|$ means concatenating the contents of two collections. The subsequent layer of the Patch Transformer takes the fusion of the previous layer's output and $g_t^l$ as its input. This is achieved by adding $g_t^l$ to the position that corresponds to $g_t^{l-1}$ in the output sequence while leaving the other parts of the output sequence unchanged. The formulation for this operation is as follows:

$$Y_{\text{in}}^l[I] = Y_{\text{out}}^{l-1}[i] = \begin{cases} Y_{\text{out}}^{l-1}[i] + g_t^l, & i = n + t(n+1), \\ & t = 0, 1, 2, \ldots, T; \\ Y_{\text{out}}^{l-1}[i], & \text{otherwise.} \end{cases} \tag{7}$$

The output feature $h_t^l := Y_{out}^l[n + t(n+1)]$ extracted from the output of the $l$-th layer of the Patch Transformer is fed into a linear head to predict the action, denoted as $\hat{a}_t = \phi(h_t^l)$ when it is the final layer. Further elaboration on this connection method is provided in Sec. 4.3.

Table 1: Results for 1% DQN-replay Atari datasets. We evaluate the performance of GDT on four Atari games using three different seeds, and report the mean and variance of the results. Best mean scores are highlighted in bold. The assessment reveals that GDT surpasses conventional RL algorithms on tasks and achieves better performance than DT across all games. In contrast, GDT-plus attains the highest average performance when compared to all baseline algorithms.

| Game | CQL | QR-DQN | REM | BC | DT | GDT | StAR | GDT-plus |
|------|-----|--------|-----|-----|-----|-----|------|----------|
| Breakout | 211.1 | 17.1 | 8.9 | $138.9 \pm 61.7$ | $267.5 \pm 97.5$ | $393.5 \pm 98.8$ | $436.1 \pm 40.0$ | $\mathbf{441.7 \pm 41.0}$ |
| Qbert | **104.2** | 0 | 0 | $17.3 \pm 14.7$ | $15.4 \pm 11.4$ | $45.5 \pm 14.6$ | $51.2 \pm 11.5$ | $51.7 \pm 20.8$ |
| Pong | **111.9** | 18 | 0.5 | $85.2 \pm 20.0$ | $106.1 \pm 8.1$ | $108.4 \pm 4.7$ | $110.8 \pm 4.8$ | $111.2 \pm 4.6$ |
| Seaquest | 1.7 | 0.4 | 0.7 | $2.1 \pm 0.3$ | $2.5 \pm 0.4$ | $\mathbf{2.8 \pm 0.1}$ | $1.7 \pm 0.3$ | $2.7 \pm 0.1$ |
| **Average** | 107.2 | 8.9 | 2.5 | 60.9 | 97.9 | 137.6 | 150.0 | **151.8** |

Table 2: Results for D4RL datasets. The performance of GDT is evaluated using three different seeds, and the mean and variance are reported. Best mean scores are highlighted in bold. The results demonstrate that GDT exhibits superior performance compared to conventional RL algorithms and sequence modeling methods, with GDT-plus achieving the highest performance.

| Dataset | Env | CQL | BEAR | IQL | BCQ | BC | DT | TT | Diffuser | GDT | StAR | GDT-plus |
|---------|-----|-----|------|-----|-----|-----|-----|-----|----------|-----|------|----------|
| M-Expert | HalfCheetah | 62.4 | 53.4 | 86.7 | 69.6 | 59.9 | 86.8 | 40.8 | 79.8 | $92.4 \pm 0.1$ | $\mathbf{93.7 \pm 0.1}$ | $93.2 \pm 0.1$ |
| M-Expert | Hopper | 111.0 | 96.3 | 91.5 | 109.1 | 79.6 | 107.6 | 106.0 | 107.2 | $110.9 \pm 0.1$ | $\mathbf{111.1 \pm 0.2}$ | $\mathbf{111.1 \pm 0.1}$ |
| M-Expert | Walker | 98.7 | 40.1 | **109.6** | 67.3 | 36.6 | 108.1 | 91.0 | 108.4 | $109.3 \pm 0.1$ | $109.0 \pm 0.1$ | $107.7 \pm 0.1$ |
| Medium | HalfCheetah | 44.4 | 41.7 | **47.4** | 41.5 | 43.1 | 42.6 | 44.0 | 44.2 | $42.9 \pm 0.1$ | $42.9 \pm 0.1$ | $42.9 \pm 0.1$ |
| Medium | Hopper | 58.0 | 52.1 | 66.3 | 65.1 | 63.9 | 67.6 | 67.4 | 58.5 | $69.5 \pm 1.8$ | $59.5 \pm 4.2$ | $\mathbf{77.1 \pm 2.5}$ |
| Medium | Walker | 79.2 | 59.1 | 78.3 | 52.0 | 77.3 | 74.0 | **81.3** | 79.7 | $77.8 \pm 0.4$ | $73.8 \pm 3.5$ | $76.5 \pm 0.7$ |
| M-Replay | HalfCheetah | **46.2** | 38.6 | 44.2 | 34.8 | 4.3 | 36.6 | 44.1 | 42.2 | $39.9 \pm 0.1$ | $36.8 \pm 3.3$ | $40.5 \pm 0.1$ |
| M-Replay | Hopper | 48.6 | 33.7 | 94.7 | 31.1 | 27.6 | 82.7 | **99.4** | 96.8 | $83.3 \pm 3.9$ | $29.2 \pm 4.3$ | $85.3 \pm 25.2$ |
| M-Replay | Walker | 26.7 | 19.2 | 73.9 | 13.7 | 36.9 | 66.6 | **79.4** | 61.2 | $74.8 \pm 1.9$ | $39.8 \pm 5.1$ | $77.5 \pm 1.3$ |
| **Average** | | 63.9 | 48.2 | 77.0 | 53.8 | 46.4 | 74.7 | 72.6 | 75.3 | 77.9 | 66.2 | **79.1** |
| Human | Pen | 37.5 | -1 | 71.5 | 66.9 | 63.9 | 79.5 | 36.4 | - | $91.7 \pm 1.7$ | $77.9 \pm 3.4$ | $\mathbf{92.5 \pm 5.1}$ |
| Human | Hammer | 4.4 | 0.3 | 1.4 | 0.9 | 1.2 | 3.1 | 0.8 | - | $3.3 \pm 0.5$ | $3.7 \pm 1.6$ | $\mathbf{5.5 \pm 1.0}$ |
| Human | Door | 9.9 | -0.3 | 4.3 | -0.05 | 2 | 14.8 | 0.1 | - | $19.5 \pm 2.2$ | $1.5 \pm 0.5$ | $\mathbf{20.6 \pm 3.1}$ |
| Human | Relocate | 0.2 | -0.3 | 0.1 | -0.04 | 0.1 | 0.3 | 0.0 | - | $\mathbf{0.7 \pm 0.2}$ | $0.1 \pm 0.1$ | $0.6 \pm 0.2$ |
| Cloned | Pen | 39.2 | 26.5 | 37.3 | 50.9 | 37 | 75.8 | 11.4 | - | $76.5 \pm 6.2$ | $33.1 \pm 3.1$ | $\mathbf{86.2 \pm 6.7}$ |
| Cloned | Hammer | 2.1 | 0.3 | 2.1 | 0.4 | 0.6 | 3.0 | 0.5 | - | $4.9 \pm 2.0$ | $0.3 \pm 0.1$ | $\mathbf{8.9 \pm 2.1}$ |
| Cloned | Door | 0.4 | -0.1 | 1.6 | 0.01 | 0.0 | 16.3 | -0.1 | - | $16.2 \pm 3.5$ | $0.0 \pm 0.1$ | $\mathbf{19.8 \pm 2.0}$ |
| Cloned | Relocate | -0.1 | -0.3 | -0.2 | -0.3 | -0.3 | 0.2 | -0.1 | - | $0.2 \pm 0.1$ | $-0.1 \pm 0.1$ | $\mathbf{0.7 \pm 0.2}$ |
| **Average** | | 11.7 | 3.1 | 14.8 | 14.8 | 13.1 | 24.1 | 6.1 | - | 26.6 | 14.6 | **29.4** |
| Complete | Kitchen | 43.8 | 0 | 62.5 | 0.8 | **65** | 50.8 | - | - | $46.1 \pm 3.1$ | $40.8 \pm 3.4$ | $43.8 \pm 2.4$ |
| Partial | Kitchen | 49.8 | 0 | 46.3 | 9.3 | 38 | 57.9 | - | - | $69.0 \pm 11.3$ | $12.3 \pm 10.2$ | $\mathbf{73.3 \pm 0.7}$ |
| **Average** | | 46.8 | 0 | 54.4 | 5.0 | 51.5 | 54.4 | - | - | 57.6 | 26.6 | **58.6** |

## 3.4 TRAINING PROCEDURE

GDT is a drop-in replacement for DT as the training and inference procedures remain the same. However, additional graph construction for the input sequence is required for GDT. Specifically, a graph $\mathcal{G} = \langle \mathcal{V}, \mathcal{E} \rangle$ is constructed from a minibatch of length $K$ (total $3K$ tokens: return-to-go, state, and action) where $\mathcal{V}$ represents the node embedding matrix, and $\mathcal{E}$ represents the adjacency matrix. The constructed graph is then input into the Graph Transformer, and attention scores are calculated using both $\mathcal{V}$ and $\mathcal{E}$. The learning objective for discrete environments can be formulated as follows:

$$\mathbb{E}_{(\hat{R},s,a)\sim\mathcal{T}} \left[ \frac{1}{T} \sum_{t=1}^{T} (a_t - \pi_{\text{GDT}}(s_{-K:t}, \hat{R}_{-K:t}, a_{-K:t-1}))^2 \right]. \quad (8)$$

To end this section, we give several comments on the proposed GDT method. Compared with DT, which uses a serialized input, GDT represents the input sequence as a graph with a causal relationship, enabling the Graph Transformer to capture dependencies between fundamentally different concepts. On the other hand, compared with methods that implicitly learn Markovian patterns (Janner et al., 2021) or introduce an additional Step Transformer (Shang et al., 2022), GDT directly incorporates the Markovian relationships in the input. This feature allows the model to handle dependencies between sequences and tokens effectively, leading to improved performance without additional computational overhead. Additionally, the introduced Patch Transformer can maintain

fine-grained spatial information using ViT-like image patches, which is particularly important in visual tasks and can improve action prediction accuracy in such environments.

## 4 EXPERIMENT

In this section, we provide a comprehensive evaluation of the proposed GDT model, which is designed to capture the complex relationships among graph-structured data and make effective decisions based on them. Our main objective is to assess the effectiveness of GDT in comparison to two popular algorithms: offline algorithms based on TD-learning and trajectory optimization algorithms. TD-learning is a widely adopted algorithm in RL due to its remarkable sampling efficiency and impressive performance on various RL tasks. On the other hand, trajectory optimization algorithms, represented by DT, have gained increasing attention in recent years due to their ability to learn from expert demonstrations and achieve performance comparable to TD-learning in various RL tasks. We conduct a comprehensive evaluation of the performance of the GDT model on a range of tasks. Specifically, we evaluate the performance of GDT on the widely used Atari benchmark (Bellemare et al., 2013), which consists of a set of discrete control tasks, as well as on the D4RL benchmark (Fu et al., 2020), which comprises a variety of continuous control tasks. Note that we refer to the model as GDT-plus when incorporating the Patch Transformer, and we categorize these methods into three groups based on the *employed approach* and the *number of parameters* to ensure a fair comparison. The detailed training parameters and MACs for our methods are provided in Appendix B.3.

### 4.1 ATARI

The Atari benchmark (Bellemare et al., 2013) is a well-recognized and widely-adopted benchmark for evaluating the performance of RL algorithms. In this study, we choose four games from the Atari benchmark, namely Breakout, Pong, Qbert, and Seaquest, each of which requires the agent to handle high-dimensional visual inputs and complex credit assignment. Similar to prior work (Chen et al., 2021), we construct the offline dataset by sampling 1% of the DQN replay buffer dataset (Agarwal et al., 2020), which consists of nearly 500k transition steps. To enable fair comparisons, we follow the normalization protocol proposed in Hafner et al. (2020), where the final scores are normalized such that a score of 100 represents the expert level performance and a score of 0 represents the performance of a random policy.

For baseline benchmark, we compare GDT with TD-learning-based algorithms, including CQL (Kumar et al., 2020), QR-DQN (Dabney et al., 2018), and REM (Agarwal et al., 2020), and several trajectory optimization algorithms, including DT (Chen et al., 2021), StARformer (Shang et al., 2022), and behavior cloning (BC), and report the results from original papers.

Table 1 presents the comparison of our proposed method with these offline baseline methods on four games. The results show that our method achieves comparable performance to CQL in three out of four games, while significantly outperforming the other methods in all four games. This indicates that our approach, which introduces the causal relationships in the input and leverages the Graph Transformer accordingly, is superior to the other methods. To ensure a fair comparison with StAR, we further introduce a Patch Transformer to incorporate fine-grained spatial information and report the results as GDT-plus. The results demonstrate that GDT-plus achieves comparable or superior performance to StAR on all four Atari games, emphasizing the significance of fine-grained information on these games. Compared with GDT, the success of the Patch Transformer in incorporating such information into action prediction is also highlighted.

### 4.2 D4RL

The D4RL benchmark (Fu et al., 2020) evaluates the performance of RL algorithms in continuous control tasks, particularly in robotic manipulation tasks with challenging control and decision-making in continuous action spaces. In this study, we focus on three standard locomotion environments from OpenAI Gym: HalfCheetah, Hopper, and Walker, utilizing three distinct dataset configurations (medium, medium-replay, and medium-expert). Additionally, we incorporate four games from the Adroit MuJoCo models, including Pen, Hammer, Door, and Relocate, using both human and cloned datasets. We also incorporate FrankaKitchen with complete and partial datasets. To ensure fair comparisons, we also normalize the scores according to the protocol established in (Fu et al., 2020), where a score of 100 corresponds to an expert policy.

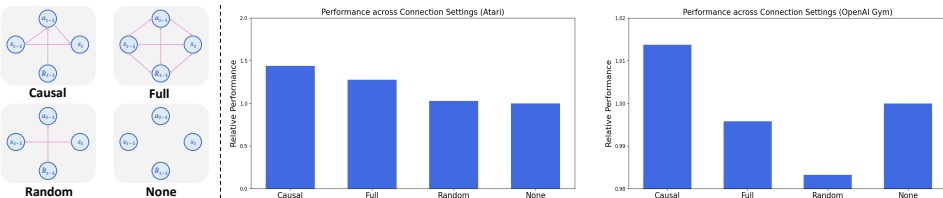

Figure 3: Comparison of different graph connection methods. The left panel illustrates four different graph connection methods: Causal, Full, Random, and None. The right panel shows the relative performance comparison of these methods.

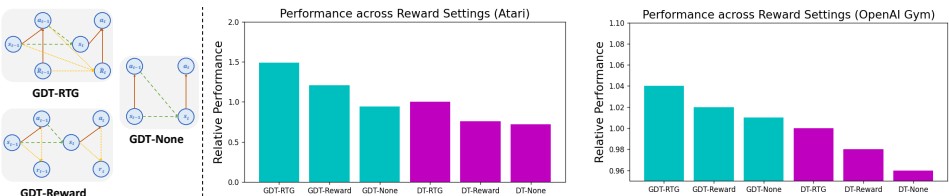

Figure 4: The left panel depicts the input graph structure of the GDT for three reward settings: return-to-go (RTG), stepwise reward (Reward), and no reward (None). The right panel displays the performance comparison between GDT and DT.

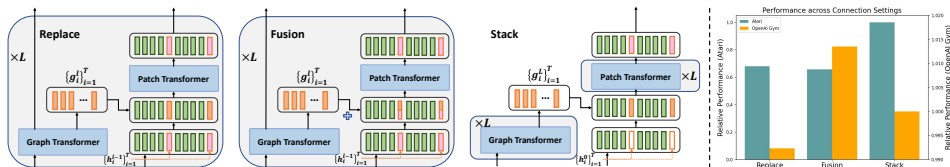

Figure 5: Performance comparison of Patch Transformer with different connection methods. The left panel illustrates the three connection methods, namely Replace, Fusion, and Stack. The right panel shows the corresponding performance comparison.

In our baseline, we conduct a comparative analysis of GDT against a set of conventional model-free methods, such as CQL (Kumar et al., 2020), BEAR (Kumar et al., 2019a), IQL (Kostrikov et al., 2021b), BCQ (Fujimoto et al., 2019a). Additionally, we include a set of trajectory optimization methods for comparison, including DT (Chen et al., 2021), TT Janner et al. (2021), Diffuser Janner et al. (2022), and StARformer (Shang et al., 2022), which represent the current strong baselines. The performance of IQL, DT, TT, and Diffuser are reported from the original papers, while the results of CQL, BCQ, and BEAR are reported from Fakoor et al. (2021), and the other methods are run by us for a fair comparison. This comprehensive evaluation encompasses a variety of techniques and conducts a thorough examination of the effectiveness of GDT in comparison to state-of-the-art algorithms. Detailed descriptions of these algorithms are provided in Appendix B.1.

The results presented in Table 2 demonstrate the superior performance of GDT on most of the evaluated tasks, with superior performance to the state-of-the-art TD-learning and trajectory optimization algorithms in all of these tasks. In the D4RL environment, the input state is represented as a vector rather than an image, as is the case in the Atari environment. Therefore, when introducing the Patch Transformer, the fine-grained spatial information is still obtained by embedding the state vector. In this context, our plug-in module continues to showcase effectiveness, as evidenced by GDT-plus consistently achieving the highest performance. Conversely, other methods that also utilize image patches exhibit a negative impact, further underscoring the distinct efficacy of our approach.

### 4.3 ABLATIONS

In comparison to the DT model, GDT transforms the input sequence into a graph with a causal relationship and feeds it into the Graph Transformer. Therefore, we initially investigate the impact

of the graph structure on the performance of GDT. Additionally, we examine the influence of reward settings, which have a greater effect on the performance of the DT model. We also explore three ways to connect the Patch Transformer, which is utilized to capture detailed spatial information. To provide a comprehensive evaluation, all ablation experiments are conducted on both Atari and OpenAI Gym (Medium dataset) environments.

**Graph Representation Setting.** As we transform the input sequence into a graph with causal relationships, we aim to investigate the impact of the graph structure on the overall performance. Given that the graph connection method has a complexity of $\mathcal{O}(n^2)$, we primarily explore four different connection methods: causal connection, full connection, random connection, and none connection. The results in Figure 3 demonstrate that the causal connection method outperforms the other methods in both environments. The full and none connection methods yield similar performances, while the random connection method is significantly impacted by the environment. Although this connection method can still achieve better results when historical information is crucial, it can be counterproductive in environments that are more reliant on causal relationships.

**Reward Setting.** In this study, we investigate the impact of reward setting on GDT's performance. Specifically, we examine the effect of three reward settings, namely return-to-go (RTG), stepwise reward (Reward), and none reward (None). RTG is a commonly used setting in sequence modeling methods (Eysenbach et al., 2020; Li et al., 2020; Srivastava et al., 2019), and its experimental results are greatly influenced by the value of the return-to-go hyper-parameter. Reward refers to the immediate reward generated by the environment after each step, which is commonly used in traditional TD-learning-based algorithms (Hessel et al., 2018; Schulman et al., 2017; Haarnoja et al., 2018). The None setting usually corresponds to straightforward behavior cloning.

The results are presented in Figure 4. It should be noted that since GDT takes a graph with a causal relationship as input, the causal relationship changes correspondingly for each reward setting, as shown in the left part of the figure. The introduction of reward has improved the performance of both methods compared to the none reward setting. However, DT is more reliant on return-to-go than GDT, and there is a large difference in performance between the two reward settings for DT. Importantly, GDT still outperforms DT with the return-to-go setting under the stepwise reward setting. This demonstrates that introducing causality in the input can reduce the dependence of sequence modeling on the return-to-go. Furthermore, under the stepwise reward setting, the graph input structure of GDT is similar to the dynamic modeling method in model-based algorithms, which is worth further research to expand and introduce in model-based approaches.

**Patch Transformer Connection Method.** To capture fine-grained spatial information, we introduce an optional Patch Transformer to improve action prediction. We denote the output variables of Graph Transformer as $g_t^l$ and the corresponding input variables in Patch Transformer as $h_t^l$. There are three possible ways to connect the two: (1) in each layer, $g_t^l$ replaces $h_t^{l-1}$ (GDT-Replace); (2) in each layer, $g_t^l$ is added to $h_t^{l-1}$ (GDT-Fusion); and (3) the last layer of $g_t^L$ is used as the initial layer $h_t^0$ (GDT-Stack). The experimental results in Figure 5 demonstrate that in the Atari environment with high information density, stacking features at the end is the most effective approach for feature refinement. In contrast, for the Gym environment with less information, feature refinement through the fusion of different abstraction levels achieves better performance.

## 5    CONCLUSION

In summary, this paper introduces the Graph Decision Transformer (GDT), an innovative offline RL methodology that transforms input sequences into causal graphs. This approach effectively captures potential dependencies between distinct concepts and enhances the learning of temporal and causal relationships. The empirical results presented demonstrate that GDT either matches or outperforms existing state-of-the-art offline RL methods across image-based Atari and D4RL benchmark tasks.

Our work highlights the potential of graph-structured inputs in RL, which has been less explored compared to other deep learning domains. We believe that our proposed GDT approach can inspire further research in RL, especially in tasks where spatial and temporal dependencies are essential, such as robotics, autonomous driving, and video games. By further investigating graph-structured inputs, the potential emerges for the development of more efficient and impactful RL algorithms, with applicability spanning diverse real-world contexts.

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

# Appendices

## A  DETAILED INPUT GRAPH DESIGN

The concept of "return-to-go" serves as a guiding factor for the agent's anticipated cumulative reward, aiming for an ideal $\hat{R}_T = 0$ at the terminal time step. Consequently, it can be construed as $\hat{R}_t = \hat{R}_{t-1} - r_{t-1}$, where $\hat{R}_t$ and $s_t$ jointly determine the selection of action $a_t$ (taking into account the current state and return-to-go to effectively reduce return-to-g to 0). Simultaneously, the current state $s_t$ and action $a_t$ establish the reward $r_t$ for the current time step and the state $s_{t+1}$ for the next time step, following the MDP formulation. This arrangement thereby shapes the expression for the succeeding time step: $\hat{R}_{t+1} = \hat{R}_t - r_t$.

## B  EXPERIMENTAL DETAILS

### B.1  BASELINE

For the Atari environment, we evaluate GDT against several state-of-the-art non-Transformer offline RL methods, such as CQLKumar et al. (2020), QR-DQNDabney et al. (2018), and REM Agarwal et al. (2020), and several imitation learning algorithms, including DT Chen et al. (2021), StARformer Shang et al. (2022), and straightforward behavior cloning. We report results from the corresponding papers for CQL, REM, and QR-DQN. For DT, there is a slight discrepancy between Chen et al. (2021) and Shang et al. (2022); we report raw data provided to us by DT authors. As StARformer performs well on Atari, we use it as the main comparison object for GDT-plus. In our behavior cloning setting, the agent lacks access to reward signals and online data from the environment, making the problem even more challenging. This differs from traditional imitation learning approaches that can collect new data and perform Inverse Reinforcement Learning Abbeel & Ng (2004); Ng et al. (2000). To create this setting, we remove the rewards from the dataset used in offline RL.

In the Gym environment, our evaluation encompasses a comprehensive comparison between GDT and an array of model-free methods, each contributing to the landscape of offline RL. This array includes established algorithms such as CQL (Kumar et al., 2020), BEAR (Kumar et al., 2019a), IQL (Kostrikov et al., 2021b), and BCQ (Fujimoto et al., 2019a). To offer a more comprehensive evaluation, we extend our scrutiny to encompass a series of sequence modeling methods that span the trajectory optimization domain. These methods encompass DT (Chen et al., 2021), TT Janner et al. (2021), Diffuser Janner et al. (2022), and StARformer (Shang et al., 2022), each of which constitutes a current formidable baseline within the realm of offline RL. For instance, DT and TT were pioneering algorithms that initially approached RL as a sequence modeling problem, harnessing transformers to attend to the input sequence and predict subsequent actions. In contrast, Diffuser treated RL as a trajectory optimization problem, leveraging a diffusion model to denoise and refine the generated trajectory for improved performance. By examining a diverse set of approaches, we strive to provide a thorough assessment of GDT's performance relative to a broad spectrum of existing methodologies. The performance of CQL, IQL, BCQ, DT, TT, and Diffuser are reported from the original papers, while the results of BEAR are reported from the D4RL paper, and the other methods are run by us for a fair comparison. Note that we re-run the experimental results on Gym for StARformer based on the official code published by the authors, as the original paper used the DeepMind Control Suite (DMC) environment with image input instead of vector input like Gym.

### B.2  TRAINING RESOURCES

We use one NVIDIA A100 GPU (SXM4) to train each model. Training each model typically takes 8-20 hours. However, since each environment needs to be trained three times with different seeds, the total training time is usually multiplied by three.

### B.3  MODEL PARAMETER

We also conducted a comprehensive comparison of our algorithm's model size to elucidate the performance improvements attributed to our graph-based input and specifically designed modules, rather than merely enhanced parameters. As demonstrated in Table 3, across various D4RL bench-

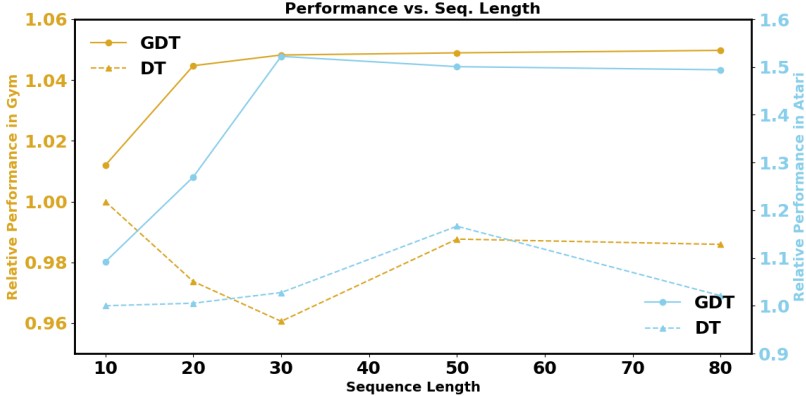

Figure 6: The performance comparison of different input sequence lengths $T \in [10, 80]$.

mark environments, GDT showcases a minute increase of only 0.003M parameters. Remarkably, this minute augmentation in parameters yields significant enhancements across Gym, Adroit, and Kitchen benchmarks. Conversely, within the context of the Atari environment, a slightly larger increment of 0.05M parameters results in substantial improvements.

Furthermore, our introduction of an additional Patch Transformer to capture fine-grained spatial information enhances GDT-plus. Notably, the parameter count of GDT-plus is notably less than that of the StAR model, while consistently attaining superior performance across D4RL and Atari environments. This outcome robustly underscores the effectiveness of our proposed approach.

Table 3: Comparison of MACs and Params.

| Method | MACs | Params | Gym | Adroit | Kitchen | MACs | Params | Atari |
|---|---|---|---|---|---|---|---|---|
| DT | 38.011M | 0.020M | 74.7 | 24.1 | 54.4 | 18.331G | 0.993M | 97.9 |
| **GDT** (ours) | 39.977M | 0.023M | 77.9 | 26.6 | 57.6 | 18.623G | 1.045M | 137.6 |
| StAR | 2.376G | 0.872M | 66.2 | 14.6 | 26.6 | 133.632G | 14.358M | 150.0 |
| **GDT-plus** (ours) | 0.324G | 0.103M | **79.1** | **29.4** | **58.6** | 29.052G | 1.145M | **151.8** |

## C  ADDITIONAL ABLATIONS

We also perform an ablation study on the length of input sequence to examine the Graph Transformer's ability to rely on long sequences. While in a Markovian environment, the state at the previous moment is often sufficient to determine the current action, the DT experiment (Chen et al., 2021) reveals that past information is valuable for the sequence modeling method in Atari environments, where longer sequences tend to yield better results than those of length 1. Subsequently, we explore the impact of different sequence lengths on performance and compare the results of DT and GDT, demonstrating the superior ability of GDT to handle long sequence inputs.

Figure 6 presents the impact of sequence length on DT and GDT performance. Our results demonstrate that the performance of GDT exhibits a continuous improvement trend until a sequence length of 50, after which it reaches a saturation point. On the other hand, DT experiences a notable decline in performance with increasing sequence lengths. By incorporating causal relationships in the input, GDT enables the Graph Transformer to effectively handle the dependencies among long sequences, leading to improved performance without additional computational overhead.

## D  HYPER-PARAMETERS

Tables 4 and 5 provide a comprehensive list of hyper-parameters for our proposed GDT and GDT-plus models applied to Atari and D4RL benchmark environments. To ensure a fair comparison, we adopt similar hyper-parameter settings to Decision-Transformer Chen et al. (2021), including the

number of Transformer layers, multi-head self-attention heads, and embedding dimensions in our Graph Transformer, as well as learning rate and optimizer configurations.

Table 4: Hyperparameters of GDT and GDT-plus for Atari experiments.

| Hyperparameter | Value |
|---|---|
| Layers (Graph Transformer) | 6 |
| MSA heads (Graph Transformer) | 8 |
| Embedding dimension (Graph Transformer) | 128 |
| Batch size | 128 |
| Context length $K$ | 50 Pong |
| | 30 Breakout, Qbert, Seaquest |
| Return-to-go conditioning | 120 Breakout |
| | 5000 Qbert |
| | 20 Pong |
| | 1450 Seaquest |
| Nonlinearity | ReLU, encoder |
| | GeLU, otherwise |
| Max epochs | 10 |
| Dropout | 0.1 |
| Learning rate | $6 * 10^{-4}$ |
| Grad norm clip | 1.0 |
| Weight decay | 0.1 |
| Warmup tokens | $512 * 20$ |
| Final tokens | $6 * 500000 * K$ |
| Connection method | Stack |
| Layers (Path Transformer) | 2 |
| Image patch size | 14 |
| MSA heads (Path Transformer) | 4 |
| Embedding dimension (Path Transformer) | 64 |

Table 5: Hyperparameters of GDT and GDT-plus for D4RL benchmark experiments.

| Hyperparameter | Value |
|---|---|
| Layers (Graph Transformer) | 3 |
| MSA heads (Graph Transformer) | 1 |
| Embedding dimension (Graph Transformer) | 128 |
| Nonlinearity function | ReLU |
| Batch size | 64 |
| Context length $K$ | 20 |
| Return-to-go conditioning | 6000 HalfCheetah |
| | 3600 Hopper |
| | 5000 Walker |
| | 12000 Pen, Hammer |
| | 2000 Door |
| | 3000 Relocate |
| | 500 Kitchen |
| Dropout | 0.1 |
| Learning rate | $10^{-4}$ |
| Grad norm clip | 0.25 |
| Weight decay | $10^{-4}$ |
| Connection method | Fusion |
| Layers (Path Transformer) | 3 |
| MSA heads (Path Transformer) | 1 |
| Embedding dimension (Path Transformer) | 256 |

