# OpenReview forum: "Graph Decision Transformer"
_ICLR.cc/2024/Conference — ICLR 2024 Conference Withdrawn Submission_

### Official Review · Reviewer_ZPDm · 2023-10-26

**Soundness:** 2 fair
**Presentation:** 3 good
**Contribution:** 2 fair
**Rating:** 5
**Confidence:** 4

**Summary:**

This paper introduces the Graph Decision Transformer (GDT) for offline Reinforcement Learning. Based on Decision Transformer, this paper changes input sequences into causal graphs and includes additional edge modeling into the original attention mechanism for better causal relationship modeling. In addition, this paper introduces a VIT-like model named Patch Transformer for better spatial information modeling. Results show that GDT performs as well or better than current offline RL methods like CQL and DT on Atari and D4RL tasks.

**Strengths:**

- The writing is easy to follow
- The paper provides rich ablation studies

**Weaknesses:**

- The effectiveness of the graph decision transformer is not so convincing. The performance of DT and GDT is quite close in D4RL (the gap is only about 3 points). Taking the performance variance into account, this is not a significant improvement. It looks like the main performance boost in GDT-plus comes from the Patch Transformer. This is not so cool as the causal relationship modeling is the main story of this project, and the results suggest that the proposed causal modeling method may not be effective in D4RL.
- The motivation of the Patch Transformer is not strong. Why do we need to introduce a new Patch Transformer for better image modeling, instead of utilizing the widely-used VIT here?

**Questions:**

- The main motivation for introducing the complex additional graph edge modeling is to model the causal relationship. However, this can be also done by simply keeping the original DT architecture and masking out the no-causal tokens in the attention maps, without adding any new modules. Have authors tried this before?
- What is “i” in Eq.7?
- In the second paragraph of Sec.3.1, the author said “The MDP is a framework where an agent is asked to make a decision based on the current state st;”. This is not a correct definition of MDP, and should be changed.
- How well does DT perform when combined with Patch Transformer?

---

### Official Review · Reviewer_nCKe · 2023-10-30

**Soundness:** 3 good
**Presentation:** 2 fair
**Contribution:** 2 fair
**Rating:** 5
**Confidence:** 3

**Summary:**

The paper presents the Graph Decision Transformer (GDT) for Reinforcement Learning (RL). Traditional RL methods, using Markov Decision Processes, face challenges in real-world applications, leading to the rise of offline RL. However, treating RL as a sequence modeling task with Transformers introduces new complications. To overcome this, GDT represents RL sequences as graph structures, effectively managing dependencies and long-term sequences. By integrating Graph and Sequence Transformers, GDT captures detailed spatial information. Experimental tests on Atari and OpenAI Gym show GDT's superior performance over existing methods.

**Strengths:**

1. This paper employs a relation-enhanced mechanism to account for temporal and causal relationships, potentially making the current decision transformer more robust and effective at capturing relations between states, rewards and rewards.

2. The path transformer is introduced to gather fine-grained spatial information for visual inputs, like the Atari environment.

**Weaknesses:**

**Major Concerns:**
1. One of the significant concerns is the computational complexity of the suggested method. The model consists of various components, each adding layers of information, such as through input concatenation. This seems to result in information redundancy rather than improving the architectural design.

2. The empirical evidence presented does not sufficiently demonstrate the efficacy of the proposed method. For instance, in the Atari games results, CQL records best scores in 2 out of 4 instances, while GDT scores the best once, and GDT-plus also once. Given that the patch transformer is specifically crafted for visual input environments, its contribution to the final results doesn't appear to be substantial.

**Minor Concerns:**
1. There are instances where figures are presented on pages different from their first mention. Notably, Figure 1, Table 1, and Table 2 are affected. It would be more reader-friendly if the authors reconsidered their placement for better clarity.
2. The "graph transformer" component lacks a comprehensive explanation, and there are no supporting citations provided.

**Questions:**

1. Kindly address the aforementioned concerns.

2. In the third paragraph of the introduction, you highlight three issues with the current DT design. I noticed the absence of citations for the second and third issues. Were these problems discussed in prior publications? If yes, please provide the necessary citations. If not, a justification for introducing these issues would be appreciated.

3. The author suggests that the proposed design can enhance long-term dependencies, yet this claim lacks both a detailed explanation and evaluative support. Could you provide further insight into this assertion?

---

### Official Review · Reviewer_o23G · 2023-10-30

**Soundness:** 3 good
**Presentation:** 2 fair
**Contribution:** 3 good
**Rating:** 5
**Confidence:** 5

**Summary:**

This paper proposes Graph Decision Transformer (GDT) a drop-in replacement for decision transformer (DT) based on graph-like embeddings for offline RL tasks. By enforcing graph-like dependencies in state, action and reward embeddings, the authors claim GDT can capture token information with high impacts compared to vanilla attending-to-all scheme. Specifically, this work employs a pre-defined dependency between state, action and reward tokens. To reflect the enforced dependency in attention module in Transformer, this work uses a modified attention score, which is done by adding edge dependencies in query and key vectors. Experimental results show GDT can achieve better performance than DT in multiple benchmarks.

**Strengths:**

- The consideration of explicit dependencies between tokens are intuitive and meaningful in modeling RL sequences, because state, action, and reward have different meanings. Close related fields like autonomous driving also use this concept in modeling different agents and objects on the road.

- The proposed method to modify query and key vectors in self-attention, under the consideration of RL-related dependencies, is somehow novel and interesting.

- The exploration on effectiveness of different kinds of dependencies are necessary. Since GDT forces the dependency, it is very informative to show the how different design choices affects the performance.

- The experimental studies on many environments are appreciated and most of them show GDT is outperforming baselines.

**Weaknesses:**

1. **Clearness of the proposed method.** The proposed method is interesting and somehow novel. However, Section 3.2 is not easy to follow.

   * I think there are some disconnections between Equations (1) (2) and actual attention mechanism, and how the proposed modification is applied to Transformer layers. Is the graph-based dependency considered every Transformer layer or is it only provided initially at the first layer?
   * Also, the $r_* \to r_*$ is learned should be clearly described.
   * The notations should be clearly defined for Equations (1-4).

2. **Soundness of proposed modified attention calculation.** I somehow understand the purpose is to fuse "extra" information, which referes to the enforced dependency we consider in RL context. Here are few questions:

   * What's the reason of only adding this to key and query, and leaving value untouched? This seems to be different than usual design of "cross-attention", which uses external query and local (or "self") key and value.
   * If the design can be supported by some experimental studies or theoratical justifications, it would be great to show its novelty and importance.
   * The modified score function considers both directions of a dependency. I'm wondering whether there is information leak that violates the temporal ordering.
   * Is the attention in Transformer layers still considering all tokens or masked according to the graph?

3. **Some comparable (baseline) methods.** The usage of graph dependency is great. In this case, an off-the-shelf graph neural network (GNN) architecture could also create this embedding. What's the main difference between the proposed way and GNNs? Can such method be used to create embeddings for GDT?

4. **Dependency designs.** The dependency is mainly RL-oriented. According to the figures, an action seems to dependent on the most recent return/reward. However, usually we model a policy $a=\pi(s)$ that does not consider it. It will be great to support this point by experiments or justifications. Essentailly this would extend the experiments summaried in Figure 3 and 4.

5. **Clarification on experimental settings.** In D4RL experiments, is GDT-plus using image inputs rather than state inputs? Then how it is compared to DT and TT for example?

**Questions:**

Please see the above weaknesses part.

---

### Official Review · Reviewer_osuH · 2023-10-31

**Soundness:** 2 fair
**Presentation:** 2 fair
**Contribution:** 2 fair
**Rating:** 3
**Confidence:** 3

**Summary:**

This paper proposes a method called Graph Decision Transformer (GDT) for offline reinforcement learning, which explicitly captures additional dependencies between adjacent states, actions, and rewards compared to a vanilla decision transformer.
In addition, this paper extends GDT with an optional Patch Transformer for better representation from visual inputs, named GDT-plus.
Experiments and ablations are conducted on Atari and D4RL and the proposed method achieves some advantages.

**Strengths:**

The paper is well-structured and easy to follow.
Experiments are conducted on two datasets and diverse tasks and consider taking advantage of visual inputs.
The hyperparameters and training details are well documented.

**Weaknesses:**

1. The intuition of the proposed graph representation is not clearly presented. Specifically, if the next state only depends on the current state and the action, which strictly follows the Markovian property, why would the current action be conditioned on both state and reward-to-go? Though the ablation on the graph representation (Fig. 3) is acknowledged, it does not strongly support the claim of the advantage of the current edge setting, considering the number of total possible settings. The authors are encouraged to expose detailed results on ablations (e.g. exact performance on each task) and explore the edge connection setting in a more generalized way (e.g. learning-based or taking advantage of evolutionary search).

2. When it comes to the architecture design, especially when merging the edge representation in a transformer layer, the authors mainly follow the architecture proposed by Cai & Lam 2020 and Yun 2019. Does this architecture benefit the learning process most? An ablation regarding graph transformer architecture is highly encouraged. Meanwhile, the contribution of this paper is more about applying a graph transformer to RL instead of proposing a new graph transformer architecture. To further eliminate confusion, the authors are encouraged to cite Cai & Lam 2020 again in the method section (3.2) and explicitly say this paper borrows the architecture design if that is the case. So that the contribution of this paper can be better highlighted.

3. In the abstract, the authors claim
> However, existing approaches that use Transformers to attend to all tokens plainly may dilute the truly essential relation priors due to information overload.

However, I could not find clear evidence to support this claim in the paper.

4. The title of this paper does not hint at the problem the paper tries to solve.

### Minor:

1. The edge connections in Fig. 1 and Fig. 3 are somehow misleading because they don't reflect the connection between the state, action current reward-to-go, and the reward-to-go at the next timestamp.


### Reference:
Deng Cai and Wai Lam. Graph transformer for graph-to-sequence learning. In AAAI, 2020.

Seongjun Yun, Minbyul Jeong, Raehyun Kim, Jaewoo Kang, and Hyunwoo J Kim. Graph transformer networks. NeurIPS, 2019.

**Questions:**

In addition to the questions and concerns mentioned in the weakness section:
1. How to project the visual inputs into tokens? How many visual frames are stacked as one state?
2. Why the depth of models are quite different in Atari and D4RL (6 vs 3 layers)? Do you have a specific reason to select the particular depth?